# Giant Cell Arteritis: Can Simple Ultrasound Examination Prevent Complex Consequences?

**DOI:** 10.3390/diagnostics14182071

**Published:** 2024-09-19

**Authors:** Wiktoria Stańska, Robert Kruszewski, Aleksandra Juszkiewicz, Artur Bachta, Witold Tłustochowicz

**Affiliations:** Department of Internal Medicine and Rheumatology, Military Institute of Medicine—National Research Institute, 04-141 Warsaw, Poland; rkruszewski@wim.mil.pl (R.K.); ajuszkiewicz@wim.mil.pl (A.J.); abachta@wim.mil.pl (A.B.); wtlustochowicz@wim.mil.pl (W.T.)

**Keywords:** GCA, vision loss, headache, ultrasonography, EULAR, OMERACT

## Abstract

Giant cell arteritis (GCA) is a rare disease of the arteries, occurring mainly in the elderly. Although the involvement of temporal arteries can be mostly symptomatic, the occlusion of ophthalmic arteries has severe consequences. GCA affecting temporal arteries is an emergency requiring quick commencement of treatment with glucocorticoids due to the serious consequences of neglect—blindness. According to the new guidelines released by EULAR, ultrasound is the tool of choice in support of the clinical diagnosis of giant cell arteritis, replacing temporal artery biopsy (TAB), as it is a sensitive and non-invasive method that is widely available. The main limitation is that the reliability of this imaging is based on the technical expertise of ultrasonographers. However, performing imaging should not delay commencing the treatment. In this work, we present ultrasound images from a case report of a 74-year-old female patient where difficulties in establishing a diagnosis led to vision loss in both eyes. In this example, we describe the ultrasound findings in giant cell arteritis, emphasizing its usefulness in supporting a diagnosis of GCA.

Giant cell arteritis (GCA) is a rare disease of the arteries, affecting large and medium vessels, with cranial arteries—branches of carotid arteries—the most frequently involved. The condition occurs mainly in the elderly population, predominantly over 50 [1]. As the pathophysiology of GCA involves myointimal proliferation, it inevitably leads to ischemic events due to vessel occlusion [1]. Because of the involvement of cranial arteries in most cases, the signs and symptoms mainly include headaches of new onset in the temples and jaw claudication. Scalp tenderness and, importantly, visual disturbances are not uncommon [2]. That is why new-onset headaches, especially in the elderly population, should alert healthcare professionals. However, there are also general non-specific symptoms, such as weight loss, fatigue, and fever, with elevated inflammatory markers [3].

On physical examination, tender and thickened temporal arteries can be found. Giant cell arteritis, if left untreated, leads to blindness in 30% to 50% of patients, and vision loss is irreversible if the steroid treatment is not administered within a few hours [4]. The awareness of the doctors who consult the patient first is crucial as the patient requires rheumatology consultation immediately to shorten the diagnostic path and implement treatment without delay [1].

A 74-year-old female patient was consulted by a family medicine doctor several times because of neck pain radiating to the temporal areas of the head. The patient had a history of hypertension and was treated with lisinopril and hydrochlorothiazide (10 + 12.5 mg/day). The pain did not respond to a short course of dexibuprofen, and the painful sensation gradually increased. Moreover, she began having headaches, night sweats, edema of the submandibular area, and trismus also occurred. The condition was treated as a common cold with amoxicillin with clavulanic acid. However, the patient started experiencing light flashes two days after administering the antibiotic. The vision symptoms persisted, and additionally, nine days later, the patient’s vision deteriorated in the right eye. The day after that, the patient reported difficulties with color vision. Eventually, the day after, blindness occurred. 

Thus, the patient was urgently admitted to the local Ophthalmology Department. The patient presented with anisocoria (L > P), the pupillary light response was medium, and she denied having any sense of light. No pursuit movements of the eyes were observed. Eventually, fundus examination revealed papilledema of the left eye and right optic nerve atrophy.

In the Ophthalmology Department, imaging was performed, including CT of the face, MRI of the orbits, and head MRI. None of them showed any significant abnormalities. Laboratory blood tests revealed an elevated CRP level of 109.8 mg/L. Upon the findings mentioned above, giant cell arteritis was suspected.

Three intravenous pulses of 1000 mg of methylprednisolone were administered on three consecutive days, then switched to oral dexamethasone at a dose of 16 mg/day. As an adjuvant therapy, vinpocetine i.v. was added. Regardless of the treatment, the vision did not improve.

Therefore, the patient was referred to the Clinical Rheumatology Department to continue the evaluations and modify the treatment. On admission, she presented with vision loss in both eyes and swelling of both temporal arteries. The erythrocyte sedimentation rate (ESR) and CRP were back to normal after initial therapy in the Ophthalmology Department, with values of 11 mm/h and 1 mg/L, respectively. There were also no significant abnormalities in other laboratory tests at that point.

In the rheumatology ward, CT angiography was carried out, and it showed a lack of contrast filling in the ophthalmic arteries. An ultrasound examination showed thickened walls of the temporal and carotid arteries with a halo sign (Figure 1B–D) and positive compression sign (Figure 2). Upon analyzing the whole clinical picture and imaging studies, the primary diagnosis of giant cell arteritis was confirmed. Therapy with glucocorticoids was continued with a daily dose tapered to 45 mg of prednisone on the day of discharge. Methotrexate at an initial dose of 20 mg per week in subcutaneous injections was added, with a planned rapid escalation to a dose of 25 mg/week. As a standard approach in methotrexate therapy, the patient was advised to take folic acid once weekly at a dose of 15 mg. The patient was also consulted by an ophthalmologist once more. The eye specialist recommended a referral to rehabilitation, aiming to improve the quality of life with blindness and enable the patient to adapt to functioning in new conditions. The patient was discharged from the hospital in a general good condition and was further evaluated in an ambulatory mode. 

The diagnostic process of GCA is challenging as some patients may present non-specific symptoms, mimicking other, less urgent conditions. Moreover, when a patient’s clinical picture suggests GCA, it requires clinical experience and expertise to establish a prompt diagnosis, as a delay leads inevitably to serious consequences such as blindness [1]. Vision deterioration is often one of the first signs of GCA, as happened in the case of our patient. Recent registrations of patients with GCA showed the possible risk factors that are associated with visual involvement, which are older age and jaw claudication, while polymyalgia rheumatica, fever, a longer duration of symptoms, and a higher ESR lowered the risk of visual symptoms [6].

Imaging is recommended for every patient suspected of GCA. The “golden standard” for diagnosing giant cell arteritis has been performing temporal artery biopsy or conventional angiography for decades. However, according to the 2023 update of the EULAR [7] recommendations, ultrasound can be the first-line imaging test to be performed in cases of the suspicion of giant cell arteritis, thanks to its high diagnostic value (pooled sensitivity from studies with low RoB of 88% and a specificity of 96%). The imaging studies aim to investigate the mural inflammatory changes in the affected arteries [7]. 

Due to the overlap of intra- and extra-cranial phenotypes of giant cell arteritis, according to the guidelines, axillary arteries should also be included in the ultrasound examination. It broadens the diagnostic yield and provides higher sensitivities than checking temporal arteries only [8]. In cases where neither temporal nor axillary artery scans are diagnostic, additional vessels can be included in the examination, such as facial, occipital, carotid, vertebral, subclavian, and femoral arteries, as well as the aorta [7].

According to European guidelines, TAB is still an adequate option, especially when there is no access to imaging or the sonographer’s expertise is insufficient [7]. The preponderance of any of these diagnostic tools is still contentious, as seen in discrepancies between European and American guidelines for giant cell arteritis [8,9]. While the EULAR guidelines are raising the predominance of ultrasound over temporal artery biopsy, the American College of Rheumatology and Vasculitis Foundation are still reserved towards comprehensive utilization of ultrasound [9]. TAB is put conditionally in the first place as an optimal diagnostic approach, with ultrasound imaging only as a complementary tool in centers with appropriate training and expertise. Such hesitancy towards ultrasound is explained by the lesser experience of rheumatologists and radiologists in the US in using ultrasound to diagnose the involvement of temporal arteries in GCA [8].

It is worth emphasizing that even though imaging is recommended as a support to establish a diagnosis, it should not delay commencing the treatment because of the severe consequences—permanent vision loss and other ischemic complications of GCA [7]. Treatment consisting of steroids should be implemented as soon as possible. It was shown that the delay in introducing steroids when visual disturbances/visual loss are already present is the strongest risk factor for permanent blindness [1]. It was shown that performing an ultrasound examination after introducing glucocorticoid treatment gives as reliable results as performing it before starting the treatment. Therefore, it is recommended that an ultrasound, if it cannot be performed before introducing treatment, should be carried out within 72 h of pharmacotherapy [7].

Ultrasound is a readily available non-invasive method for quickly diagnosing giant cell arteritis. This is compared to angio-CT, which has several drawbacks, such as it requires an imaging center in a hospital, the procedure takes more time, radiation exposure occurs, and the patient may exhibit contraindications for administering a contrast agent. Moreover, the waiting time for a CT scan may be too long for patients with conditions requiring rapid action, such as giant cell arteritis [7]. Regardless of the undisputable advantage of ultrasound over other imaging methods, it has some limitations. It requires trained and well-qualified practitioners whose experience in ultrasonography of head and neck vessels can diagnose the pathologies with confidence and a low risk of error [7].

All patients presenting typical signs and symptoms of GCA and raised inflammatory markers (C-reactive protein, Erythrocyte Sedimentation Rate) should be promptly sent to a specialistic center with state-of-the-art imaging and TAB for a work-up of a multidisciplinary team consisting of a rheumatologist, radiologist, ophthalmologist, and/or neurologist [1,7,8,9]. It is essential that the diagnostic process is prompt and without any unnecessary delay so the patient can receive the proper treatment on time by the appropriate team of healthcare professionals. Prolonged diagnosis, and thereby delayed treatment, results in irreversible consequences [1]. That is why the awareness and proper training of doctors who consult the patient first—at a low level of the healthcare ladder—is important, so they can refer patients to fast-track clinics that are properly prepared for such emergencies [1,7,8,9].

Due to the non-specific nature of the symptoms, the diagnosis of GCA is difficult. That is why this pathology is considered to be underdiagnosed, because of the low degree of clinical suspicion [9]. Moreover, as patients with GCA are referred to the hospital from Primary Care or out-of-hospital emergencies, it implies the need for comprehensive protocols that enable bidirectional communication channels guaranteeing referrals between specialists and the continuity of the care between the facilities. This could coordinate the services involved in the diagnostic and treatment process [9]. Another solution for improving the quality of care of patients with GCA is developing training sessions to continue the clinical education for healthcare professionals, at all levels of care involved in the process of the management of GCA [9].

The diagnosis of GCA is established upon the clinical picture and high inflammatory markers, supported by imaging or temporal artery biopsy. Giant cell arteritis with affected temporal arteries is an emergency, requiring immediate intervention and prompt administration of high doses of glucocorticosteroids. In these cases, the prognosis depends on the healthcare professionals, as a quick response and proper management can save many lives every year [1].

## Figures and Tables

**Figure 1 diagnostics-14-02071-f001:**
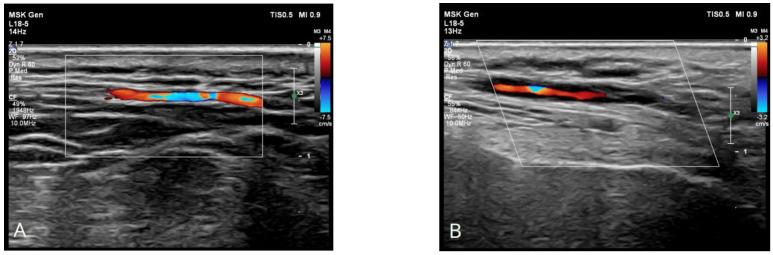
Ultrasonographic imaging of the temporal artery. Comparison of ultrasound images of an unchanged temporal artery (**A**) and inflammation of the temporal artery in giant cell arteritis (**B**). Both longitudinal (**C**) and transverse (**D**) views of the inflamed artery reveal the narrowing of the arterial lumen and homogenous, hypoechoic wall thickening (white markers), well delineated towards the luminal side, known as the “halo sign” [5].

**Figure 2 diagnostics-14-02071-f002:**
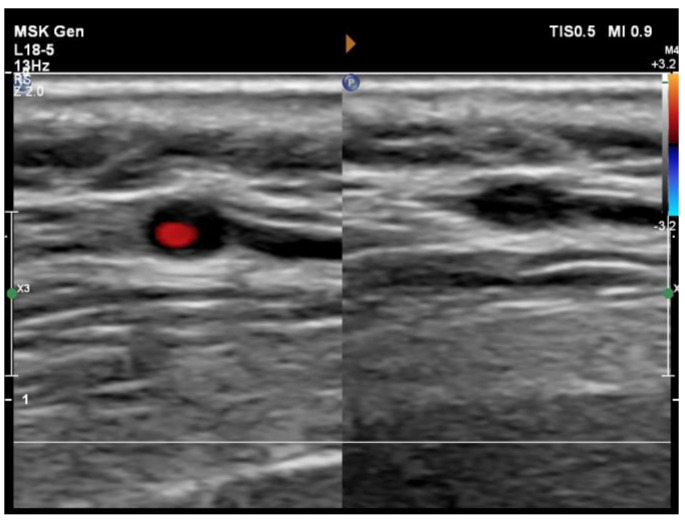
Detection of arterial inflammation by “compression sign”. Pressure applied with an ultrasound probe collapses the artery. The resulting restriction of blood flow is confirmed by the absence or limitation of a Doppler signal in the arterial lumen. However, the thickened inflamed arterial wall remains visible as a hypoechoic area contrasting with the mid-echogenic to hyperechoic surrounding tissue [5].

## Data Availability

No new data were created or analyzed in this study.

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
