# Peer review of "Giant Cell Arteritis: Can Simple Ultrasound Examination Prevent Complex Consequences?"

_diagnostics, 2024, doi:10.3390/diagnostics14182071_

Round 1

Reviewer 1 Report

Comments and Suggestions for Authors

Giant Cell Arteritis: Can Simple Ultrasound Examination Prevent Complex Consequences?

Comment

Manuscript

Line

Page

It is better to use easily and simple language

Worth emphasising is that performing imaging

18

1

It would be better if the authors added short details about the case like age and sex and the main presentation.

we present ultrasound images from a case report of a patient where

20

1

It is better to follow the scientific rules of writing abbreviations.

Giant cell arteritis is a rare disease of the

27

1

Many sentences and paragraphs are without citing references

To the readers, it would be better to divide the manuscript to multiple sections starting with introduction ,case presentation ,and lastly discussion

Duration and type of the used drugs

The pain did not respond to nonsteroidal anti-inflammatory drugs

49

2

Give duration

And investigations done to the patient in this period.

However, the patient started 52 experiencing light flashes two days after administering the antibi

52

2

No investigations were mentioned until eye complain. I think this is not appropriate.

Many details are missing.

The authors did not mention for how long time the patient received pulse methyl prednisolone.

Three intravenous pulses of 1000 mg of methylprednisolone were administered

67

2

In general, oral long acting steroids are not preferred in practice

The authors did not mention the equivalent dose of steroids received.

oral dexamethasone in a dose of 16 mg/day

86

2

Many details are lost as the follow up levels of laboratory data.

Erythrocyte sedimentation rate (ESR) and CRP were back to normal

74

2

It is not clear why the authors chose dexamethasone and if this is an equivalent dose or the actual dose of dexamethasone

decided to up dexamethasone to 45 mg/day

81

3

This is also the maximum dose of methotrexate which is not recommended to start with in this age

methotrexate in a dose of 25 mg per week

83

3

Why the authors did not start with the standard dose from the start.

As a standard approach in methotrexate therapy, the patient was advised to take 83 folic acid once weekly in a dose of 15 mg.

Why ...rehabilitation for what.

What did this sentence add to the report of the case?

The patient was also consulted by an 84 ophthalmologist once more, recommending a referral to rehabilitation

85

3

The authors again did not follow the scientific rules for writing abbreviations

giant cell arteritis

112

4

No one can deny the role of imaging in GCA.

So, I think the authors should focus also on the missing diagnosis on discussion .

Comments on the Quality of English Language

moderate 

Author Response

Dear Sir or Madam,

We express our sincere appreciation for reviewing our manuscript. We believe that your invaluable assessment let us improve the article and make it more clear for the readers. Please find below our responses to your kind comments.

Page 1 – line 18: “It is better to use easily and simple language”

We simplified some more complex sentences.

Page 1- line 20: “It would be better if the authors added short details about the case like age and sex and the main presentation.”

We have already described the demographic parameters of the patient in the main text. However, upon your comment, we added them in the abstract as well.

Page 2-line 27: “It is better to follow the scientific rules of writing abbreviations.”

Thank you for your comment. We corrected that.

„Many sentences and paragraphs are without citing references”

We filled the gaps in citing references.

„To the readers, it would be better to divide the manuscript to multiple sections starting with introduction, case presentation, and lastly discussion.”

We agree with the layout of the manuscript suggested by you. However, for publishing in the Topical Section “Interesting Images” we received precise guidelines for refraining from a “regular (introduction, methods, results, discussion) structure.” Therefore, if dividing the main text into sections is allowed by the Editor of “Interesting Images” we would be more than happy to correct it.

Page 2 – line 49: “Duration and type of the used drugs”

The NSAID, dexibuprofen, was prescribed by the GP, before admitting to the hospital. Therefore we have no medical documentation regarding the duration of treatment. However, given the following onset of the vision symptoms and ceasing NSAID, it was a short course of treatment with dexibuprofen.

Page 2-line 52: “Give duration and investigations done to the patient in this period.” “No investigations were mentioned until eye complain. I think this is not appropriate.”

The lack of extensive investigations during this period was because the beginning of the evaluations took place at the lowest level of healthcare – Primary Care. Therefore, the inaccuracies are due to it. The general practitioner performed a physical examination and ultrasound of the salivary glands, unfortunately, the result of which is not known to us, because we did not receive medical documentation of the patient from the GP office. We rely solely on the patient’s oral report. According to the patient no other diagnostics tests were carried out at that time. Moreover, according to the patient, the result of the ultrasound was deemed by the practitioner as inconclusive.

The described perceived shortcomings are one of the reasons to share this case report, with a perspective of a need to change the evaluations and sensitize the readers to the consequences of the delayed diagnosis and importantly what caused the delay in the first place.

Page 2 – line 67: “Many details are missing. The authors did not mention for how long time the patient received pulse methyl prednisolone.”

We added the missing information to the manuscript body. The patient received intravenous methylprednisolone for three consecutive days.

Page 2- line 86: “In general, oral long acting steroids are not preferred in practice. The authors did not mention the equivalent dose of steroids received.”

Dexamethasone was used briefly, only in the Local Ophthalmology Department, until the patient was referred to the Clinical Rheumatology Department. According to the recommendations, steroids are used in the initial dose of 40-60 mg/day of prednisolone equivalent. It does not exclude using the equivalent dose of dexamethasone. However, the preferable treatment should indeed be prednisone, with a planned tappering of dosage, which took place (45 mg of prednisone, not dexamethasone as we originally mentioned). That is why it is so crucial that the patient is diagnosed quickly and referred to the rheumatologist, to receive a proper treatment. We emphasized it in the discussion.

Page 2 – line 74: “Many details are lost as the follow up levels of laboratory data.”

Concerning the laboratory findings, the most essential are the parameters of the inflammation, which we presented as far as they were available to us. At the moment of admitting the patient to the Rheumatology Department, the parameters were already normalised (CRP 1 mg/l, ESR 11 mm/h), due to the steroid treatment that had been administered in the Ophthalmology Department. The only aberration in the blood tests was leukocytosis, an effect of the steroid treatment. Other findings of the blood tests were unremarkable.

Page 3- line 81: “It is not clear why the authors chose dexamethasone and if this is an equivalent dose or the actual dose of dexamethasone”

Thank you for catching this error. While writing the manuscript, there must have been a mix-up with the drug name. Indeed, the patient received prednisone at an actual dose of 45 mg/day.

Page 3-line 83: “This is also the maximum dose of methotrexate which is not recommended to start with in this age.”

Initially, the patient received the methotrexate at a dose of 20 mg/week with good tolerance. That was the actual dose at the moment of discharging the patient from the hospital. Albeit, it was rapidly upped to 25 mg/week after discharge. Routinely, in our clinic we start the treatment with a dose of 15 mg per week, however in this case we aimed to achieve the therapeutic level quickly, to be able to lower the dose of glucocorticosteroids.

“Why the authors did not start with the standard dose from the start”

Regarding the folic acid supplementation during the methotrexate therapy, there are no precise recommendations. Usually, the dose of folic acid oscillates around 5-15 mg/week. In Poland, we routinely use the dose of 15 mg per week.

Page 3-line 85: “Why ...rehabilitation for what. What did this sentence add to the report of the case?”

The rehabilitation was aimed at improving the quality of life with blindness and enabling the patient to adapt to function in new conditions.

Page 4 – line 112: “The authors again did not follow the scientific rules for writing abbreviations”

Thank you for noticing it. We corrected it.

“No one can deny the role of imaging in GCA. So, I think the authors should focus also on the missing diagnosis on discussion .”

Thank you for your review. We had already mentioned that in our manuscript because it is one of the main objectives of writing up this case report. However, we elaborated on this more in the discussion.

Reviewer 2 Report

Comments and Suggestions for Authors

This paper discusses a clinical case of GCA typical of extra-cranial phenotype that loses vision due to a delay in the diagnostic process.

Main concerns:

1. I think that the fatal consequences in this patient were caused by a delay in diagnosis and in the start of treatment. There are many approaches to optimize the management of this disease, but it is always necessary to have a multidisciplinary vision and use a defined algorithm shared with other disciplines, such as those described in some recent publications, v.gr.: PMID: 38142973, which should be taken into account in the discussion of the case.

2. The authors do not specify possible risk factors associated with visual loss in this case. In a recent registry of patients with GCA, some risk factors related to the visual manifestations of this entity are mentioned, such as age and jaw claudication as independent predictors of visual manifestations, whereas polymyalgia rheumatica, longer symptom duration, and high ESR reduce the risk of visual involvement (PMID: 38244610). We recommend to discuss this work.

3. Why was dexamethasone started after IV methylprednisolone boluses instead of oral prednisolone? Also, I think this patient was a good candidate for tocilizumab at the start of therapy along with oral prednisone and 6-MP boluses due to vision loss (PMID: 35329914 // PMID: 36377586). I think this aspect should be justified in the discussion and comment on the content of this case report.

Author Response

Dear Sir or Madam,

Sincerely thank you for your kind reviews that let us make our manuscript better. Please find below the responses to your comments.

Ad. 1

Thank you for your review. We had already mentioned the need for a quick referral to the specialistic centers and we perceive an incessant education of the healthcare professional as essential. Nevertheless, we elaborated on this in the discussion, citing the works suggested by you.

Ad. 2

Thank you, we filled this gap in the manuscript.

Ad. 3

Dexamethasone was used briefly, only in the Local Ophthalmology Department, until the patient was referred to the Clinical Rheumatology Department. According to the recommendations, steroids are used in the initial dose of 40-60 mg/day of prednisolone equivalent. It does not exclude using the equivalent dose of dexamethasone. However, the preferable treatment should indeed be prednisone, with a planned tapering of dosage, which took place (45 mg of prednisone, not dexamethasone as we originally mentioned). That is why it is so crucial that the patient is diagnosed quickly and referred to the rheumatologist, to receive a proper treatment. We emphasized it in the discussion.

According to the EULAR 2018 and ACR 2021 recommendations the patient should receive, besides steroids, tocilizumab or methotrexate, as an alternative. As we wrote in the manuscript, upon analyzing the whole clinical picture the decision of administering methotrexate as a first-line treatment was made.

Round 2

Reviewer 1 Report

Comments and Suggestions for Authors

The manuscript improved 

Comments on the Quality of English Language

minor

Reviewer 2 Report

Comments and Suggestions for Authors

The authors have made an effort to improve their manuscript in order its publication. 

Comments on the Quality of English Language

I think the manuscript needs a review by a native editor.